# WorldForge: Unlocking Emergent 3D/4D Generation in Video Diffusion Model via Training-Free Guidance

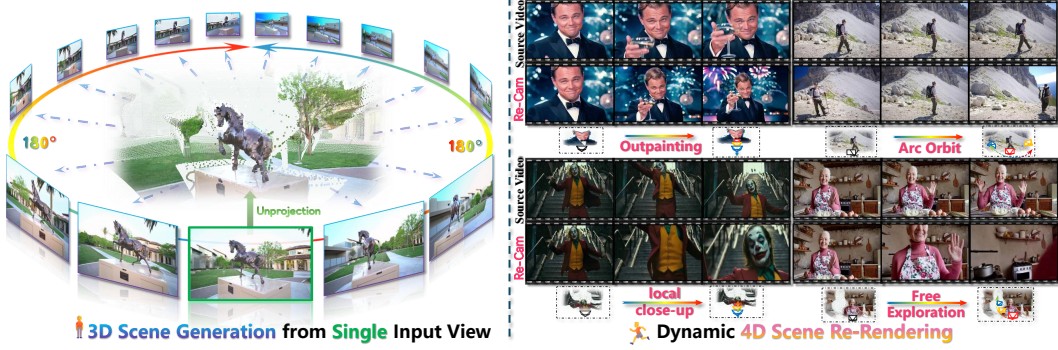

Figure 1: We present WorldForge, a fully training-free framework leveraging a pre-trained video diffusion model for various 3D/4D tasks, such as monocular 3D scene generation (left) and dynamic 4D scene re-rendering (right), enabling precise camera trajectory control and high-quality outputs.

## Abstract

Recent video diffusion models show immense potential for spatial intelligence tasks due to their rich world priors, but this is undermined by limited controllability, poor spatial-temporal consistency, and entangled scene-camera dynamics. Existing solutions, such as model fine-tuning and warping-based repainting, struggle with scalability, generalization, and robustness against artifacts. To address this, we propose WorldForge, a training-free, inference-time framework composed of three tightly coupled modules. 1) Intra-Step Recursive Refinement injects fine-grained trajectory guidance at denoising steps through a recursive correction loop, ensuring motion remains aligned with the target path. 2) Flow-Gated Latent Fusion leverages optical flow similarity to decouple motion from appearance in the latent space and selectively inject trajectory guidance into motion-related channels. 3) Dual-Path Self-Corrective Guidance compares guided and unguided denoising paths to adaptively correct trajectory drift caused by noisy or misaligned structural signals. Together, these components inject fine-grained, trajectory-aligned guidance without training, achieving both accurate motion control and photorealistic content generation. Our framework is plug-and-play and model-agnostic, enabling broad applicability across various 3D/4D tasks. Extensive experiments demonstrate that our method achieves state-of-the-art performance in trajectory adherence, geometric consistency, and perceptual quality, outperforming both training-intensive and inference-only baselines.

## 1 Introduction

Recent video diffusion models (VDMs) (Blattmann et al., 2023; Wan et al., 2025; Yang et al., 2024; Google DeepMind, 2025) have significantly advanced spatial intelligence (Cao et al., 2025) tasks like 3D/4D understanding (Bahmani et al., 2025a;b), reconstruction (Wang et al., 2025a; Wu et al., 2025; Shi et al., 2024), and generation (Yu et al., 2024c; 2025). Trained on vast video datasets, these

models encode rich spatiotemporal priors, enabling realistic spatial transformations for applications like novel view synthesis (You et al., 2025; Xiao et al., 2025), panoramic video (Wang et al., 2024b; Ma et al., 2024a), and dynamic scene reconstruction (Bai et al., 2025a; Yu et al., 2025; Van Hoorick et al., 2024). Furthermore, VDMs are increasingly used to build "world models" (Bar et al., 2025; Duan et al., 2025; Bruce et al., 2024), which are structured internal representations that support predictive reasoning in embodied AI.

Despite their strong priors, VDMs face fundamental limitations, including limited controllability, spatial–temporal consistency, and geometric fidelity, particularly when applied to 3D or 4D tasks (Wang et al., 2024c; He et al., 2024; Ling et al., 2024; Xing et al., 2024). They struggle to follow precise motion constraints, such as a 6-DoF camera trajectory (Hu, 2024; Ma et al., 2024b), which undermines spatial consistency in tasks such as novel view synthesis and trajectory control. These models also entangle scene and camera motion, causing unintended object deformations and scene instability when viewpoint changes are desired (Yu et al., 2024c; Liu et al., 2024). These limitations hinder their use in applications requiring structured spatial reasoning or controllable generation.

To handle these issues, prior works (Jeong et al., 2025; Ren et al., 2025; Yu et al., 2025; Zhang et al., 2025) have pursued two main directions. The first, fine-tuning on motion-conditioned data (Bai et al., 2025a; Xiao et al., 2024; Bai et al., 2025b), can improve control but is computationally costly, generalizes poorly, and risks degrading pretrained priors. The second, a "warping-and-repainting" strategy (Ma et al., 2025b; Liu et al., 2025; Ma et al., 2025a; You et al., 2025), re-projects frames along a new camera path and uses a generative model to fill occlusions. Although this approach is more flexible, it lacks robustness because pretrained models handle warped, out-of-distribution (OOD) (Yu et al., 2024a) inputs poorly, often producing artifacts and fragmented geometry. Moreover, a bias toward dynamic training data can cause hallucinated motion in static scenes. Consequently, balancing fine-grained controllability with generation quality and generalization remains a challenging open problem.

To address this challenge, we aim to inject precise control into VDMs while preserving their valuable priors. For this purpose, we propose a general inference-time guidance paradigm that leverages the rich priors of VDMs in spatial intelligence tasks, such as geometry-aware 3D scene generation and video trajectory control. Our method uses a warping-and-repainting pipeline, in which input frames are warped along a reference trajectory and then used as conditional inputs in the repainting stage. Building on this, we develop a unified, training-free framework composed of three complementary mechanisms, each designed to address a specific challenge in trajectory-controlled generation.

First, to ensure the generated motion follows the target trajectory derived from depth-based rendering (Wang et al., 2025b; Piccinelli et al., 2024), we introduce **Intra-Step Recursive Refinement (IRR)**. It embeds a micro-scale predict–correct loop within each denoising step: before the next timestep, predicted content in observed regions is replaced with the corresponding ground-truth (GT) observations. This incremental correction allows trajectory control signals to be injected at every step, enabling fine-grained aligned with the target trajectory.

Second, we observe that different channels of the VAE-encoded (Kingma & Welling, 2013; Foti et al., 2022) latent representation encode different information, with some channels specializing in appearance and others in motion. Directly overwriting all channels when injecting trajectory signals can inadvertently degrade visual details. To address this, we propose **Flow-Gated Latent Fusion (FLF)**, which leverages optical-flow similarity to selectively inject trajectory information into channels highly correlated with motion, while leaving appearance-relevant channels unmodified. This process effectively decouples appearance from motion, allowing for precise viewpoint manipulation while preserving content fidelity.

Finally, while warping-based rendering effectively enforces user-defined trajectories, it inevitably introduces noise and visual artifacts stemming from imperfect depth, occlusions, or misalignments (You et al., 2025). To balance control with quality, we propose **Dual-Path Self-Corrective Guidance (DSG)**. Inspired by CFG (Ho & Salimans, 2021), DSG uses two parallel denoising paths during inference: A non-guided path that relies on the model's priors to produce high-fidelity but uncontrolled results, and a guided path that follows the warped trajectory, ensuring camera control but risking artifacts. At each step, DSG computes the difference between these paths to create a dynamic correction term. This term adjusts the guided path toward the higher perceptual quality of

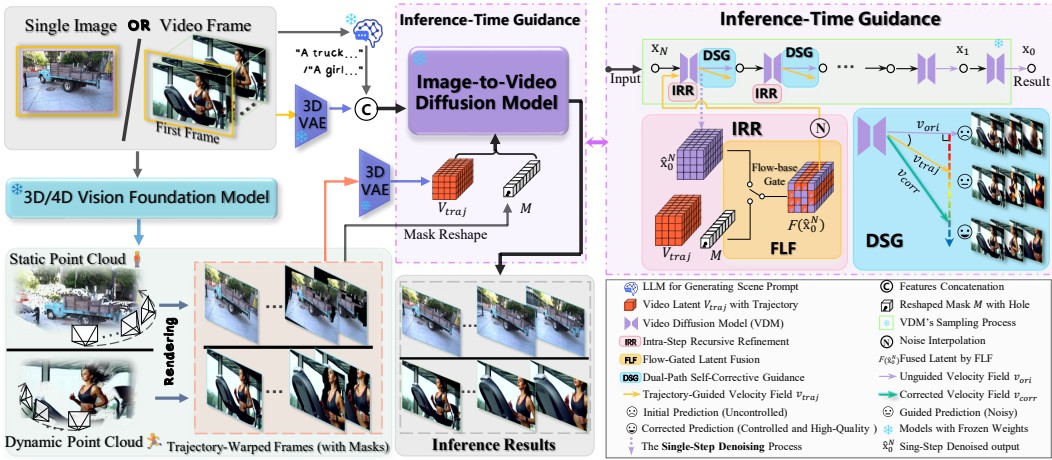

Figure 2: Overview of our porposed method. Given a single image or video frames, a vision foundation model reconstructs a scene point cloud, which is warped and rendered along a user-specified trajectory to produce a guidance video. The input image (or first frame) is also converted into a textual prompt and latent representation for an image-to-video diffusion model. Trajectory control is injected through a training-free strategy comprising IRR, FLF, and DSG (detailed in Sec. 3.2–3.4), enabling precise control and high-quality synthesis without additional training.

the non-guided path. This self-corrective mechanism mitigates artifacts from the warped trajectory while maintaining camera control, improving the video's overall structure and visual quality.

Together, these three mechanisms form a cohesive inference-time guidance framework for robust and precise trajectory control while preserving VDM priors. Our method is training-free and plug-and-play, enabling broad applicability across tasks without model retraining. It is also model-agnostic and readily adapts to backbones such as Wan 2.1 (Wan et al., 2025) and SVD (Blattmann et al., 2023). Comprehensive experiments on multiple tasks and benchmarks confirm that our approach achieves state-of-the-art (SOTA) results, improving trajectory adherence, geometric consistency, and perceptual quality over leading baselines. Our main contributions are:

- A novel, training-free paradigm for leveraging VDM priors in spatial intelligence tasks, enabling precise and stable 3D/4D trajectory control without retraining or fine-tuning.
- A synergistic inference-time guidance framework integrating **I**ntra-Step **R**ecursive **R**efinement (IRR) and **F**low-Gated **L**atent **F**usion (FLF), achieving accurate trajectory adherence while disentangling motion from content.
- **D**ual-Path **S**elf-Corrective **G**uidance (DSG), a self-referential correction mechanism that enhances spatial alignment and perceptual fidelity without auxiliary networks or retraining.
- Extensive experiments on diverse datasets and tasks show our approach achieves SOTA controllability and visual quality, even compared to training-intensive pipelines.

## 2 RELATED WORKS

We review prior work in three relevant areas: 3D static scene generation, 4D trajectory-controlled video generation, and guidance strategies for generative models.

**3D Static Scene Generation.** While 3D reconstruction (Mildenhall et al., 2020; Kerbl et al., 2023; Song et al., 2024; Gao et al., 2024; Yu et al., 2024d; Müller et al., 2022; Yao et al., 2018) and object generation (Poole et al., 2023; Wei et al., 2024; Xiang et al., 2025; Kwak et al., 2024) are advanced, they often lack scene-level priors. VDM (Blattmann et al., 2023; Wan et al., 2025; Kong et al., 2024) provide these priors and are leveraged by decoding scenes from images (Liang et al., 2025), fine-tuning on warped inputs (Yu et al., 2024c; Ma et al., 2025a), or embedding camera parameters (Wang et al., 2024c; Xiao et al., 2025). Unlike costly fine-tuning which can corrupt priors, training-free strategies (You et al., 2025; Liu et al., 2024) preserve them but must ensure geometric coherence. Our work takes this training-free approach to enhance both consistency and control.

**Trajectory-Controlled Dynamic Video Generation.** One paradigm for controllable video synthesis is to fine-tune lightweight adapters (Ma et al., 2025b; Mou et al., 2024; Yu et al., 2024b; Wang et al., 2024c) like LoRA (Hu et al., 2022) or ControlNet (Zhang et al., 2023) on video-trajectory data, conditioning on diverse inputs (Bai et al., 2025a; Yu et al., 2025; Van Hoorick et al., 2024; Gu et al., 2025). Another is the *warp-and-repaint* strategy (Ma et al., 2025b; Liu et al., 2025; Huang et al., 2025; Tian et al., 2025), which projects and inpaints frames but is prone to artifacts from noisy warps (You et al., 2025). Our work uses inference-time guidance to directly steer the diffusion process for precise motion control.

**Guidance and Control for Generative Models.** Guidance strategies steer diffusion models toward desired outputs. While Classifier-Free Guidance (CFG) (Ho & Salimans, 2021) is common, high weights can cause artifacts. More advanced techniques use auxiliary models (Karras et al., 2024; Hyung et al., 2025; Xu et al., 2023) or iterative refinement (Bai et al., 2025c) to improve sampling. In 3D/4D synthesis, guidance is used to enforce viewpoint consistency, but warp-based methods often suffer from noise-induced artifacts (Cai et al., 2024; Wang et al., 2024a). To address this, we propose DSG. It derives a correction signal from the difference between guided and unguided predictions at each step, enhancing trajectory adherence and stability without retraining.

## 3 PROPOSED METHODS

We propose an inference-time guidance strategy to balance controllability with visual fidelity for VDMs in 3D/4D tasks. Our method is a training-free framework that steers a pretrained model along a user-defined trajectory while preserving its generative priors. As shown in Fig. 2, our framework has three key components. First, Intra-Step Recursive Refinement (IRR) injects trajectory guidance from observed regions at each denoising step for consistent control (Sec. 3.2). Second, Flow-Gated Latent Fusion (FLF) decouples motion from appearance in the latent space to prevent content drift and preserve fidelity (Sec. 3.3). Finally, Dual-Path Self-Corrective Guidance (DSG) uses the difference between guided and unguided paths as a corrective signal to suppress artifacts and improve stability (Sec. 3.4). Together, these modules enable fine-grained trajectory control and unlock the model's latent 3D/4D awareness without any retraining.

### 3.1 PRELIMINARIES

Before detailing our method, we introduce the necessary preliminaries: diffusion models, guidance strategies, and trajectory-controlled video synthesis.

#### 3.1.1 DENOISING DIFFUSION MODELS AND GUIDANCE

**Diffusion Solvers.** Generative models are largely diffusion (Ho et al., 2020) or flow-based (Lipman et al., 2022). Under the SDE view, diffusion models have a deterministic ODE limit that connects to flow models via reparameterization (Gao et al., 2025) (The detailed derivation is provided in Appendix A). We use the popular DDIM sampler (Song et al., 2020a) as an example to illustrate the sampling process: it recovers the clean sample $\mathbf{x}_0$ by reversing the forward noising of a Gaussian prior $\mathbf{x}_T$. Given a noise-prediction network $\boldsymbol{\epsilon}_\theta(\mathbf{x}_t, t)$, the sampler estimates an intermediate signal $\hat{\mathbf{x}}_0$ from the current state $\mathbf{x}_t$:

$$\hat{\mathbf{x}}_0(\mathbf{x}_t, t) = \frac{\mathbf{x}_t - \sqrt{1 - \bar{\alpha}_t}\, \boldsymbol{\epsilon}_\theta(\mathbf{x}_t, t)}{\sqrt{\bar{\alpha}_t}}, \tag{1}$$

where $\bar{\alpha}_t$ denotes cumulative noise attenuation. The term $\hat{\mathbf{x}}_0(\mathbf{x}_t, t)$ is a key intermediate variable: at each step, it is the one-step denoised estimate from $\boldsymbol{\epsilon}_\theta$, evolving from a coarse prediction to a sharp final output. The next sample $\mathbf{x}_{t-1}$ is then obtained by blending $\hat{\mathbf{x}}_0$ with the predicted noise $\boldsymbol{\epsilon}_\theta$:

$$\mathbf{x}_{t-1} = \sqrt{\bar{\alpha}_{t-1}}\, \hat{\mathbf{x}}_0(\mathbf{x}_t, t) + \sqrt{1 - \bar{\alpha}_{t-1}}\, \boldsymbol{\epsilon}_\theta(\mathbf{x}_t, t). \tag{2}$$

Iterating from $t = T$ to $t = 0$ yields the final sample $\mathbf{x}_0$. Our method intervenes at this stage by *modifying $\hat{\mathbf{x}}_0$ to enforce trajectory control*. Notably, other popular solvers, such as UniPC (Zhao et al., 2023), EDM (Karras et al., 2022), and PNDM (Liu et al., 2022a), also compute $\hat{\mathbf{x}}_0$ directly or can recover it via a parameterized transformation, so our framework is broadly compatible. Since our experiments primarily use the flow-based Wan2.1 (Wan et al., 2025) model, we will subsequently detail our algorithm using a flow-based formulation.

**Classifier Free Guidance.** To improve fidelity to the condition, CFG (Ho & Salimans, 2021) adjusts the network prediction during sampling as:

$$\tilde{\epsilon}_\theta(\mathbf{x}_t, t) = \epsilon_\theta(\mathbf{x}_t, t, \mathbf{c}) + \omega_{\mathrm{CFG}} \cdot [\epsilon_\theta(\mathbf{x}_t, t, \mathbf{c}) - \epsilon_\theta(\mathbf{x}_t, t, \phi)], \tag{3}$$

where $\omega_{\mathrm{CFG}}$ is the guidance weight, with $\mathbf{c}$ and $\phi$ denoting the conditional and unconditional inputs, respectively. This interpolates conditional and unconditional scores to steer the sampling trajectory. Our approach extends this principle through a self-referential guidance mechanism that dynamically adjusts the guided prediction using the model's own unguided output at each step.

### 3.1.2 TRAJECTORY CONTROL VIA DEPTH-BASED WARPING

Our framework controls trajectories using a depth-based warping-and-repainting strategy. First, a depth-prediction network estimates camera poses and depth maps $(\mathbf{P}_q, \mathbf{D}_q)$ from single image $\mathbf{I}$ or image sequence $\{\mathbf{I}_q\}_{q=1}^{N}$ via a function $f : \{\mathbf{I}_q\}_{q=1}^{N} \to \{\mathbf{P}_q, \mathbf{D}_q\}$. Next, a warping operator $\mathcal{W}$ uses these estimates to project a source frame $\mathbf{I}_{src}$ with depth $\mathbf{D}_{src}$ from pose $\mathbf{P}_{src}$ to a target pose $\mathbf{P}_{tar}$. This yields a partial target view $\mathbf{I}'_{tar}$ and a validity mask $\mathbf{M}_{tar}$ indicating visible pixels:

$$(\mathbf{I}'_{tar}, \mathbf{M}_{tar}) = \mathcal{W}(\mathbf{I}_{src}, \mathbf{D}_{src}, \mathbf{P}_{src}, \mathbf{P}_{tar}). \tag{4}$$

The resulting warped frames and masks guide the VDMs along the target poses $\mathbf{P}_{tar}$. This guidance is limited to regions visible in the source views. With these preliminaries, we use this trajectory control to guide video generation in VDMs.

### 3.2 INTRA-STEP RECURSIVE REFINEMENT

To enable precise trajectory injection during VDM's inference processing, we introduce Intra-Step Recursive Refinement (IRR). As noted in Sec. 3.1.1, the denoising process produces an intermediate variable $\hat{\mathbf{x}}_0^{(t)}$, a coarse estimate of the final output and the baseline for later steps, where $t$ denotes the current timestep. IRR modifies $\hat{\mathbf{x}}_0^{(t)}$ to impose trajectory constraints, ensuring that generation follows the desired path.

IRR operates within the updates of Eq. (1) and Eq. (2). Given the one-step denoised sample $\hat{\mathbf{x}}_0^{(t)}$ from Eq. (1), we fuse it with the trajectory latent $\mathbf{Z}_{\mathrm{traj}}$, obtained by encoding the warped frames of Eq. (4) into latent space. We then add Gaussian noise $\epsilon$ to obtain the modified latent $\mathbf{x}'_t$:

$$\mathbf{x}'_t = (1 - w(\sigma)) \, \mathbf{F}(\hat{\mathbf{x}}_0^{(t)}, \mathbf{Z}_{\mathrm{traj}}) + w(\sigma) \cdot \epsilon, \tag{5}$$

where $\mathbf{F}(\hat{\mathbf{x}}_0^{(t)}, \mathbf{Z}_{\mathrm{traj}}) = \mathbf{M} \cdot \mathbf{Z}_{\mathrm{traj}} + (1 - \mathbf{M}) \cdot \hat{\mathbf{x}}_0^{(t)}$ copies observable warped content from $\mathbf{Z}_{\mathrm{traj}}$ into the corresponding locations of $\hat{\mathbf{x}}_0^{(t)}$ using the binary mask $\mathbf{M}$ from Eq. (4); and $\epsilon = \mathbf{x}_T \sim \mathcal{N}(\mathbf{0}, \mathbf{I})$ is a randomly sampled Gaussian noise used to re-noise the fused latent $\mathbf{F}(\hat{\mathbf{x}}_0^{(t)}, \mathbf{Z}_{\mathrm{traj}})$ so that it re-enters the denoising schedule with the injected trajectory. The re-noising is controlled by a scheduler $w(\sigma)$, and the specific strategy can differ for various diffusion and flow models. The re-noised latent $\mathbf{x}'_t$ is then fed into the network $\epsilon_\theta$, replacing the original $\mathbf{x}_t$ in Eq. (1) and Eq. (2) for the next sampling step. In summary, IRR embeds a micro predictor–corrector at each denoising step. By updating $\hat{\mathbf{x}}_0^{(t)}$ with explicit trajectory cues, it continually corrects the sampling path and ensures that synthesis follows the target trajectory precisely.

### 3.3 FLOW-GATED LATENT FUSION

In the IRR process, overwriting all latent channels with trajectory information degrades visual quality because VAE latents encode both motion and appearance. The indiscriminate update in Eq. (5) injects noise into appearance-focused channels. To address this, we propose **Flow-Gated Latent Fusion (FLF)**, a method that identifies and updates latent channels with high motion relevance.

To select motion-related channels, we use an optical-flow-based scoring scheme since flow directly reflects inter-frame motion. First, for each channel $c$ of the latent $\hat{\mathbf{x}}_0^{(t)}$ at timestep $t$, we compute the optical flow between consecutive frames to get a predicted flow $\mathcal{F}_{\mathrm{pred}}^{(t,c)}$. The resulting map for each channel has a shape of $[2, \tau, H, W]$ (flow vectors, frame pairs, spatial dimensions). Second, we

compute a GT reference flow $\mathcal{F}_{\text{gt}}^{(t,c)}$ from the target trajectory latent $\mathbf{Z}_{\text{traj}}$ in the same frame-by-frame manner. Finally, we compare the two flows within the visible regions defined by the mask $\mathbf{M}^{(c)}$.

The comparison relies on three popular optical flow metrics (Teed & Deng, 2020): Masked End-point Error (M-EPE), which measures the Euclidean distance between predicted and GT flow vectors; Masked Angular Error (M-AE), which measures their angular difference; and Outlier Percentage (Fl-all), which calculates the fraction of unreliable pixels. We combine the normalized metrics to calculate a motion similarity score $S^{(t,c)}$ for each channel in each timestep $t$:

$$S^{(t,c)} = \sum_{k \in \{\text{E, A, F}\}} \gamma_k \left( 1 - \text{Norm}_k^{(t,c)} \right), \tag{6}$$

where $\text{Norm}_k^{(t,c)} \in [0,1]$ are the normalized errors from M-EPE, M-AE, and Fl-all, and $\gamma_k$ are the weighting factors. The detailed calculation and normalization methods for these metrics are provided in the Appendix C. Higher $S^{(t,c)}$ means better flow alignment and stronger motion evidence.

To adaptively set the motion similarity threshold for each scene, we select motion-relevant channels using a dynamic threshold $\delta^{(t)} = \mu_S^{(t)} - \lambda^{(t)} \sigma_S^{(t)}$, where $\mu_S^{(t)}$ and $\sigma_S^{(t)}$ are the mean and standard deviation of all channel scores at step $t$. We schedule $\lambda^{(t)}$ to create a loose-to-tight selection over time. This matches the generative process, where early steps define broad structures and later steps handle fine details. Consequently, we guide more channels initially for structural integrity and fewer channels toward the end to preserve high-fidelity details.

Finally, the latent update selectively fuses the trajectory information $\mathbf{Z}_{\text{traj}}$ into the selected high motion-relevance channels:

$$\mathbf{FLF}(\hat{\mathbf{x}}_0^{(t)}, \mathbf{Z}_{\text{traj}}) = \begin{cases} \mathbf{M}^{(c)} \cdot \mathbf{Z}_{\text{traj}}^{(c)} + \left(1 - \mathbf{M}^{(c)}\right) \cdot \hat{\mathbf{x}}_0^{(t,c)}, & \text{if } S^{(t,c)} \geq \delta^{(t)} \\ \hat{\mathbf{x}}_0^{(t,c)}, & \text{otherwise.} \end{cases} \tag{7}$$

This FLF operator replaces the global update in Eq. (5), resulting in a more precise fusion rule: This FLF operator replaces the operator $\mathbf{F}$ in Eq. (5), resulting in a more precise fusion rule.

In summary, FLF provides fine-grained trajectory control while preserving model priors and synthesis quality. Unlike methods that restart the sampling schedule (Xu et al., 2023) or globally update the entire latent (Liu et al., 2024), FLF integrates with our IRR framework to apply selective, per-step guidance, effectively decoupling motion and appearance for precise control.

### 3.4 DUAL-PATH SELF-CORRECTIVE GUIDANCE

Trajectory latents $\mathbf{Z}_{\text{traj}}$ obtained by warping often contain distortions from depth errors, occlusions, or misalignments, which degrades synthesis quality. To mitigate this, we draw inspiration from CFG (Ho & Salimans, 2021). Conceptually, in the context of a flow model, CFG treats the unconditional prediction as a "bad" direction $\mathbf{v}_{\text{uncon}}$ and the conditional one as a "good" direction $\mathbf{v}_{\text{con}}$ (Karras et al., 2024). It then finds a "better" direction by pushing the "good" one away from the "bad" one. Based on this idea, we propose **Dual-Path Self-Corrective Guidance (DSG)**. At each iteration, IRR produces two velocity fields. The unguided velocity $\mathbf{v}_t^{\text{ori}}$ (from the original latent $\mathbf{x}_t$) stays on the data manifold with high fidelity but ignores the trajectory, which we consider a "bad" direction for control. The guided velocity $\mathbf{v}_t^{\text{traj}}$ (from the corrected latent $\mathbf{x}_t'$) may be noisy but follows the trajectory, which we consider a "good" direction. DSG uses the difference between them to find a "better" path.

However, we empirically find the difference between our $\mathbf{v}_t^{\text{traj}}$ and $\mathbf{v}_t^{\text{ori}}$ is far greater than that between the $\mathbf{v}_{\text{con}}$ and $\mathbf{v}_{\text{uncon}}$ in standard CFG. In extensive tests, we observed that the cosine similarity $\alpha_t = (\mathbf{v}_t^{\text{traj}} \cdot \mathbf{v}_t^{\text{ori}})/(\|\mathbf{v}_t^{\text{traj}}\| \cdot \|\mathbf{v}_t^{\text{ori}}\|)$ between our two paths is typically between 0.3–0.6 (an angle of 50°–70°). In contrast, the cosine similarity between $\mathbf{v}_{\text{con}}$ and $\mathbf{v}_{\text{uncon}}$ is nearly 1 (an angle close to 0°), an observation consistent with reports by Wan et al. (2025). Therefore, directly applying the CFG formula fails in our case. To address this large angular difference, we modify the guidance formula to only use the component of the "good" direction that is orthogonal to the "bad" direction. This is achieved by projecting $\mathbf{v}_t^{\text{traj}}$ onto $\mathbf{v}_t^{\text{ori}}$ (after normalizing them to equal length) and taking the difference, which avoids the adverse effects of their large directional divergence:

$$\mathbf{v}_t^{\text{corr}} = \mathbf{v}_t^{\text{traj}} + \rho \cdot \beta_t \left( \mathbf{v}_t^{\text{traj}} - \alpha_t \cdot \mathbf{v}_t^{\text{ori}} \right), \tag{8}$$

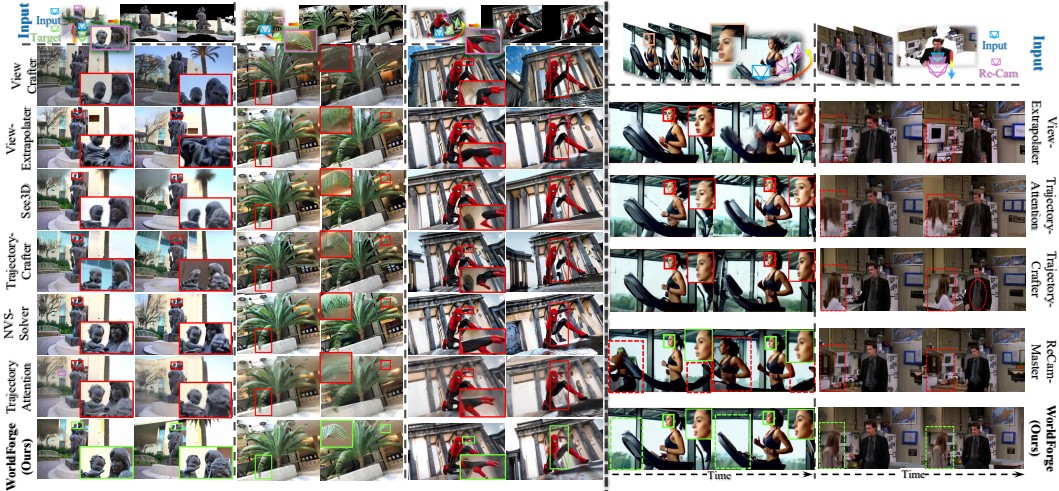

Figure 3: Qualitative comparison for 3D novel view synthesis and 4D trajectory-controlled re-rendering. (**Left**) For 3D scene generation from a single image, our method produces more consistent and plausible content compared to SOTA baselines. (**Right**) For 4D re-rendering, our approach leverages model priors to avoid common artifacts like floating heads and flattened faces, yielding more realistic results. Overall, our training-free guidance demonstrates superior performance in both static and dynamic scenes. Zoom in for more details.

where $\rho$ controls guidance strength, $\alpha_t$ is the cosine similarity, and $\beta_t = \sqrt{1 - \alpha_t^2}$ is its sine counterpart. The role of $\beta_t$ is to adaptively scale the guidance: it amplifies the correction when the paths diverge (low $\alpha_t$, high $\beta_t$) and reduces it when they agree, preserving the model's natural prediction.

In essence, DSG robustly balances precise trajectory control with high visual fidelity. Its adaptive cosine-weighting is crucial for handling the large angular difference between the guided and unguided paths. It suppresses artifacts by applying strong corrections when the paths diverge, while preserving the model's natural predictions when they align. This ensures the final corrected velocity, $\mathbf{v}_t^{\mathrm{corr}}$, steers the sample along the target motion path while maintaining the quality of the model's priors. As shown in Appendix D, DSG's visual results are far superior to naive CFG applications.

## 4 EXPERIMENTS

In this section, we present a comprehensive evaluation of our proposed training-free framework. We first outline the implementation details in Sec. 4.1. Subsequently, we demonstrate the performance of our method on 3D scene generation and 4D trajectory control in Sec. 4.2. Finally, we conduct a series of ablation studies in Sec. 4.3 to validate the effectiveness of each component of our approach. More results and details are provided in Appendix E.

### 4.1 IMPLEMENTATION DETAILS

Our framework is a training-free method that steers pre-trained VDMs for precise camera control, with no additional training or fine-tuning. Compared with pre-trained VDMs, our method incurs no training cost, while the inference time increases by approximately 40–50%, mainly due to the IRR module (see Appendix E.1 for detailed efficiency analysis).

**Setup.** Experiments primarily use the Wan2.1 Image-to-Video (I2V-14B) model (Wan et al., 2025). Generation runs on a single GPU with ≥69 GB VRAM, producing videos up to 1280×720. The per-pass sequence length depends on the chosen VDM's capacity; longer videos are obtained by concatenation. For ablation and fair comparison, we also evaluate SVD (Blattmann et al., 2023), which runs on a 24 GB RTX 4090 for 25-frame inference. VDM ablation details are in Appendix E.2.

Our pipeline follows a warp-and-repaint design. For warping, we test several depth estimation models, including VGGT (Wang et al., 2025b), UniDepth (Piccinelli et al., 2024), Mega-SaM (Li et al., 2025), and DepthCrafter (Hu et al., 2025). The method adapts well across these choices, benefiting from the VDM's strong world priors; see Appendix E.3 for depth-estimator ablations.

**Test Datasets and Metrics.** For single-view 3D scene generation, we use data from LLFF (Mildenhall et al., 2019), Tanks and Temples (Knapitsch et al., 2017), MipNeRF 360 (Barron et al., 2022), and diverse internet, real-world, and AI-generated images. We report FID (Heusel et al., 2017) and $CLIP_{sim}$ (Radford et al., 2021). For 4D trajectory control, we use challenging real-world videos, reporting FVD (Unterthiner et al., 2018) and CLIP-$V_{sim}$. For both tasks, trajectory accuracy is measured with Absolute Trajectory Error (ATE), Relative Pose Error—Translation (RPE-T), and Relative Pose Error—Rotation (RPE-R). Full metric definitions are in Appendix B.

## 4.2 3D AND 4D TRAJECTORY-CONTROLLED GENERATION

We compare our method against state-of-the-art baselines on both 3D static scene generation and 4D dynamic video control. For 3D novel view synthesis, we evaluate against both training-based (Yu et al., 2024c; 2025; Xiao et al., 2025; Ma et al., 2025a) and training-free (You et al., 2025; Liu et al., 2024) methods. For 4D trajectory control, baselines include ReCamMaster (Bai et al., 2025a) and others (Yu et al., 2025; Xiao et al., 2025; Liu et al., 2024).

Under identical evaluation settings, our training-free method consistently achieves superior results. On 3D static scenes, it outperforms both training-based and training-free baselines on public datasets (Fig. 3, Table 1). On 4D clips with challenging camera paths (e.g., arcs, dolly zooms), it yields higher visual fidelity, tighter trajectory alignment, and more coherent scene completion, matching or surpassing costly training-based approaches. In both settings, our method plausibly reconstructs unseen regions where baselines often produce distortions.

Our approach particularly excels in difficult cases. It handles human-centric scenes, which require high consistency, and can synthesize photorealistic and structurally consistent $360°$ views from a single input (Appendix E.5). By preserving model priors, it strikes a strong balance between controllability and fidelity. We tested our method on a lighter SVD model (Blattmann et al., 2023) and still achieved high visual quality, confirming its strong performance across different model scales (Appendix E.2).

Beyond benchmarks, the framework serves as a versatile tool for video post-production. It can stabilize videos by smoothing camera motion, control paths for localized super-resolution or outpainting, and perform masked edits like object addition/removal or subject try-on effects. These capabilities highlight its utility for real-world video re-rendering (more results in Appendix E.4).

Table 1: Quantitative comparison with SOTA methods on 3D static and 4D dynamic scenes, using public and internet data. We evaluate generation quality (FID, $CLIP_{sim}$ for static; FVD, CLIP-$V_{sim}$ for dynamic) and trajectory accuracy (ATE, RPE-T, RPE-R). All methods use official code with identical inputs. ↑: Higher is better, ↓: Lower is better. Our method achieves the **best** or second-best results on all metrics. Metric details are in Appendix B.

| | Generation Quality | | | | Trajectory Accuracy | | | | | |
| | Static | | Dynamic | | Static | | | Dynamic | | |
| | FID ↓ | $CLIP_{sim}$ ↑ | FVD ↓ | CLIP-$V_{sim}$ ↑ | ATE ↓ | RPE-T ↓ | RPE-R ↓ | ATE ↓ | RPE-T ↓ | RPE-R ↓ |
|---|---|---|---|---|---|---|---|---|---|---|
| See3D (Ma et al., 2025a) | 123.26 | 0.941 | – | – | 0.091 | 0.089 | 0.250 | – | – | – |
| ViewCrafter (Yu et al., 2024c) | 117.50 | 0.930 | – | – | 0.236 | 0.315 | 0.728 | – | – | – |
| ViewExtrapolator (Liu et al., 2024) | 125.50 | 0.930 | 108.48 | 0.913 | 0.183 | 0.260 | 0.882 | 1.040 | 1.208 | 4.750 |
| TrajectoryAttention (Xiao et al., 2025) | 122.37 | 0.920 | 106.94 | 0.911 | 0.159 | 0.238 | 0.532 | 0.605 | 1.238 | 3.560 |
| TrajectoryCrafter (Yu et al., 2025) | 111.49 | 0.910 | 97.31 | 0.923 | 0.090 | 0.152 | 0.267 | **0.431** | 1.078 | 8.950 |
| NVS-Solver (You et al., 2025) | 118.64 | 0.937 | – | – | 0.224 | 0.268 | 1.056 | – | – | – |
| **WorldForge (Ours)** | **96.08** | **0.948** | **93.17** | **0.938** | **0.077** | **0.086** | **0.221** | 0.527 | **0.826** | **2.690** |

## 4.3 ABLATION EXPERIMENTS

We ablate each module and design choice in our framework.

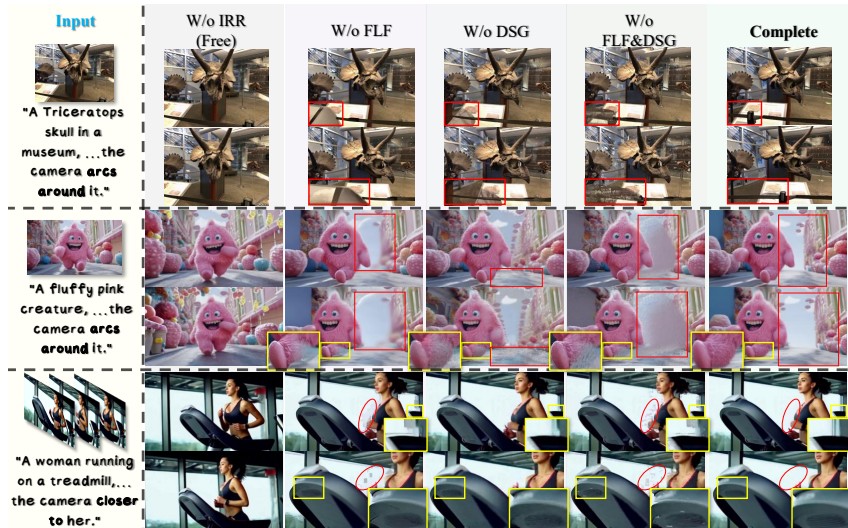

Figure 4: Ablation of the proposed components. IRR enables trajectory injection; without it, the model defaults to prompt-only free generation, and FLF/DSG cannot be applied. FLF decouples trajectory cues from noisy content; removing it introduces noise from warped frames. DSG guides sampling toward high-quality, trajectory-consistent results; without it, detail and plausibility drop. The full model achieves the best fidelity and control, demonstrating their complementary effects.

**Component Analysis.** We remove IRR, FLF, and DSG in turn (Fig. 4). Removing IRR disables trajectory guidance at inference time, resulting in failure to follow the target path. Without FLF, i.e., lacking motion/appearance separation, model priors become entangled, leading to unnatural outputs. Removing DSG introduces noise from warped trajectories into the generation process, causing artifacts and degrading visual quality. The complete model yields the best results, showing that all components are essential and work synergistically to enable robust and precise control.

**Video Model and Depth Model.** We replace Wan 2.1 (Wan et al., 2025) with the U-Net–based SVD (Blattmann et al., 2023) to test model-agnosticism. It consistently achieved strong results, demonstrating its effectiveness across a range of model architectures(Appendix E.2). Similarly, our method remains robust regardless of the warping model. We experimented with different depth estimators (Wang et al., 2025b; Li et al., 2025; Piccinelli et al., 2024; Hu et al., 2025) and found that the VDM's strong 3D priors effectively mitigate many warping artifacts. This allows for a plug-and-play integration with various depth models (Appendix E.3).

## 5 CONCLUSION

We present WorldForge, a training-free framework for trajectory-controllable generation in static 3D and dynamic 4D scenes. Our method effectively balances visual quality, generalization, and precise control in video synthesis. At its heart is a unified, inference-time guidance strategy—comprising Intra-Step Recursive Refinement (IRR), Flow-Gated Latent Fusion (FLF), and Dual-Path Self-Corrective Guidance (DSG). By decoupling motion from appearance and correcting trajectory drift, our framework injects fine-grained control while preserving the rich world priors of the base model. Extensive experiments show state-of-the-art performance on both 3D and 4D generation tasks, offering a new path for exploring spatial intelligence in large-scale generative systems.

While our framework corrects many warping-induced distortions, it can fail with severely inaccurate depth estimations (*e.g.*, completely flattened subjects or severe foreground–background entanglement). Furthermore, the global nature of our guidance offers limited control over small objects and fine details. Future work will focus on integrating fine-grained control mechanisms and applying our method to more powerful generative models.

## ETHICS AND REPRODUCIBILITY STATEMENTS

**Ethics Statement** Our method is specifically designed for 3D/4D controllable content generation via video diffusion models. The framework operates by manipulating the latent space of a pre-trained model based only on the data available in the user-provided input images or videos. As such, no additional information regarding human subjects or potentially harmful insights is introduced during the process. This approach prioritizes privacy and ethical considerations by not requiring any sensitive or external information.

**Reproducibility Statement** We are committed to the reproducibility of our work. All of our experiments are conducted in a training-free manner, which not only simplifies implementation but also enhances reproducibility by removing dependencies on training data and hardware setups. To allow for full verification of our results and to support future research, we will make our source code publicly available after it has been prepared for release.

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

## A  PROOF OF THE EQUIVALENCE BETWEEN DIFFUSION AND FLOW MODELS

We consider Flow Matching (Lipman et al., 2022; Liu et al., 2022b) as a special case of diffusion modeling (Kingma & Gao, 2023; Gao et al., 2025). In the following, we will first outline the formulation of diffusion models and then substitute the specific parameterization of Flow Matching to demonstrate their compatibility.

Given a random variable $\mathbf{x}_0$ drawn from an unknown data distribution $q_0(\mathbf{x}_0)$, a Diffusion Probabilistic Model (DPM) (Ho et al., 2020; Song et al., 2020b; Lu et al., 2022) defines a forward process that gradually transforms the data into a simple prior distribution, typically a Gaussian distribution. The conditional distribution of the noised variable $\mathbf{x}_t$ at time $t$ given the initial data $\mathbf{x}_0$ is defined as a Gaussian transition kernel (Kingma et al., 2021):

$$q_t(\mathbf{x}_t|\mathbf{x}_0) = \mathcal{N}(\mathbf{x}_t|\alpha_t\mathbf{x}_0, \sigma_t^2\mathbf{I}). \tag{9}$$

Equivalently, a sample $\mathbf{x}_t$ at any time $t \in [0, T]$ can be expressed through a reparameterization (Kingma et al., 2021; Gao et al., 2025):

$$\mathbf{x}_t = \alpha_t\mathbf{x}_0 + \sigma_t\boldsymbol{\epsilon}, \ \boldsymbol{\epsilon} \sim \mathcal{N}(\mathbf{0}, \mathbf{I}). \tag{10}$$

Here, $\alpha_t$ and $\sigma_t$ are scalar functions of time, known as the noise schedule, that control the signal-to-noise ratio. Typically, $\alpha_t$ decreases over time while $\sigma_t$ increases, satisfying a condition such as $\alpha_t^2 + \sigma_t^2 = 1$ in Variance Preserving (VP) SDEs (Ho et al., 2020; Song et al., 2020b). Kingma et al. (2021) proves that the following stochastic differential equation (SDE) has the same transition distribution in Eq. (9) for any $t \in [0, T]$:

$$d\mathbf{x}_t = f(t)\mathbf{x}_t dt + g(t)d\mathbf{w}_t, \ \mathbf{x}_0 \sim q_0(\mathbf{x}_0), \tag{11}$$

where $\mathbf{w}_t$ is a standard Wiener process. The drift coefficient $f(t)$ and the diffusion coefficient $g(t)$ can be derived using schedule parameters $\alpha_t$ and $\sigma_t$ (Kingma et al., 2021):

$$f(t) = \frac{d\log\alpha_t}{dt}, \quad g^2(t) = \frac{d\sigma_t^2}{dt} - 2\frac{d\log\alpha_t}{dt}\sigma_t^2. \tag{12}$$

The generative process of diffusion models involves reversing this forward process. This can be achieved via a corresponding reverse-time SDE (Song et al., 2020b). For more efficient generation, one can utilize the associated probability flow ordinary differential equation (PF-ODE), which shares the same marginal distributions as at each time $t$ as that of the SDE (Song et al., 2020b). This PF-ODE is given by:

$$\frac{d\mathbf{x}_t}{dt} = f(t)\mathbf{x}_t - \frac{1}{2}g^2(t)\nabla_{\mathbf{x}_t}\log p_t(\mathbf{x}_t). \tag{13}$$

By relating the score function $\nabla_{\mathbf{x}_t}\log p_t(\mathbf{x}_t)$ to the noise term via $\nabla_{\mathbf{x}_t}\log p_t(\mathbf{x}_t) \approx -\frac{\boldsymbol{\epsilon}_\theta(\mathbf{x}_t,t)}{\sigma_t}$, where $\boldsymbol{\epsilon}_\theta$ is a neural network trained to predict the noise, the ODE becomes (Karras et al., 2022; Zhao et al., 2023):

$$\frac{d\mathbf{x}_t}{dt} = f(t)\mathbf{x}_t + \frac{g^2(t)}{2\sigma_t}\boldsymbol{\epsilon}_\theta(\mathbf{x}_t, t). \tag{14}$$

Now, let us consider the forward process in Flow Matching (Lipman et al., 2022; Liu et al., 2022b). The path from a data point $\mathbf{x}_0$ to a noise sample $\boldsymbol{\epsilon}$ is defined by a simple linear interpolation:

$$\mathbf{x}_t = (1-t)\mathbf{x}_0 + t\cdot\boldsymbol{\epsilon}, \ \boldsymbol{\epsilon} \sim \mathcal{N}(\mathbf{0}, \mathbf{I}), \tag{15}$$

where $t \in [0, 1]$. By comparing Eq. (15) with the general form of the diffusion forward process in Eq. (10), we can establish a direct correspondence by setting the diffusion schedule parameters as:

$$\alpha_t = 1 - t \quad \text{and} \quad \sigma_t = t.$$

Substituting this specific parameterization into the definitions for $f(t)$ and $g(t)$ in Eq. (12), we derive the corresponding coefficients for this Flow Matching SDE:

$$f_{\text{FM}}(t) = \frac{d\log(1-t)}{dt} = \frac{-1}{1-t}, \tag{16}$$

$$g_{\text{FM}}^2(t) = \frac{d(t^2)}{dt} - 2\frac{-1}{1-t}t^2 = \frac{2t}{1-t}. \tag{17}$$

Next, we insert these specific coefficients $f_{\text{FM}}(t)$ and $g_{\text{FM}}^2(t)$ into the PF-ODE formulation from Eq. (14). To analyze the underlying dynamics, we consider the ideal case where the score is perfectly known, which is equivalent to replacing the model prediction $\epsilon_\theta(\mathbf{x}_t, t)$ with the ground-truth noise $\epsilon$. This yields:

$$
\begin{aligned}
\frac{\mathrm{d}\mathbf{x}_t}{\mathrm{d}t} &= f_{\text{FM}}(t)\mathbf{x}_t + \frac{g_{\text{FM}}^2(t)}{2\sigma_t}\epsilon \\
&= \frac{-1}{1-t}\mathbf{x}_t + \frac{2t}{2t \cdot (1-t)}\epsilon \\
&= \frac{\epsilon - \mathbf{x}_t}{1-t} \\
&= \frac{\epsilon - [(1-t)\mathbf{x}_0 + t \cdot \epsilon]}{1-t} \\
&= \frac{(1-t)\epsilon - (1-t)\mathbf{x}_0}{1-t} \\
&= \epsilon - \mathbf{x}_0.
\end{aligned}
\tag{18}
$$

This resultant vector field, $\frac{\mathrm{d}\mathbf{x}_t}{\mathrm{d}t} = \epsilon - \mathbf{x}_0$, is precisely the time derivative of the Flow Matching path defined in Eq. (15). This equivalence demonstrates that the process prescribed by Flow Matching is a specific instance of the diffusion models, corresponding to the linear noise schedule $\alpha_t = 1 - t$ and $\sigma_t = t$. Therefore, Flow Matching can be formally viewed as a subset of the broader diffusion modeling framework (Kingma & Gao, 2023; Gao et al., 2025).

## B EVALUATION METRICS

We employ seven complementary metrics to comprehensively evaluate video generation quality: FID and $\text{CLIP}_{\text{sim}}$ similarity for static scenes, FVD and $\text{CLIP-V}_{\text{sim}}$ for dynamic scenes, and ATE, RPE-T, and RPE-R for camera trajectory consistency. These metrics provide objective quantitative assessment across multiple dimensions including image realism, semantic consistency, temporal coherence, and camera motion fidelity.

### B.1 STATIC SCENE EVALUATION

**Fréchet Inception Distance (FID).** FID (Heusel et al., 2017) measures image generation quality by comparing the distribution of real and generated images in the Inception-V3 feature space. We use an ImageNet-pretrained Inception-V3 (Szegedy et al., 2016) model and extract 2048-dimensional features from the pool3 layer. The FID score is computed as:

$$
\text{FID} = \|\mu_r - \mu_g\|^2 + \text{Tr}(\Sigma_r + \Sigma_g - 2(\Sigma_r \Sigma_g)^{1/2})
\tag{19}
$$

where $\mu_r$ and $\mu_g$ are the mean vectors of real and generated image features, and $\Sigma_r$ and $\Sigma_g$ are the corresponding covariance matrices.

**CLIP Similarity.** CLIP similarity (Radford et al., 2021) evaluates the semantic similarity between generated and real images using vision-language pre-trained representations. We employ the CLIP ViT-B/32 model trained on 400 million image-text pairs. The similarity score is calculated as:

$$
\text{CLIP}_{\text{sim}} = \frac{1}{N} \sum_{i=1}^{N} \cos(f_{r,i}, f_{g,i})
\tag{20}
$$

where $f_{r,i}$ and $f_{g,i}$ are the L2-normalized 512-dimensional CLIP features of the $i$-th real and generated image pair.

### B.2 DYNAMIC SCENE EVALUATION

**Fréchet Video Distance (FVD).** FVD (Unterthiner et al., 2018) measures distributional differences between real and generated video using pretrained spatio-temporal features. We use an I3D (Inflated 3D ConvNet) pretrained on Kinetics (Carreira & Zisserman, 2017) and extract 1024-D features

from the global average pooling layer for each video clip. Following FID, we compute the Fréchet distance between the Gaussian fits of real and generated I3D features:

$$\text{FVD} = \|\mu_r - \mu_g\|_2^2 + \text{Tr}\left(\Sigma_r + \Sigma_g - 2(\Sigma_r \Sigma_g)^{1/2}\right), \tag{21}$$

with $\mu_r, \mu_g$ and $\Sigma_r, \Sigma_g$ estimated over clip-level I3D features.

**Video CLIP Similarity** (CLIP-$\text{V}_{\text{sim}}$). CLIP-$\text{V}_{\text{sim}}$ extends CLIP similarity to the temporal domain by computing frame-level semantic consistency between generated and real videos. The score is calculated as:

$$\text{CLIP-V}_{\text{sim}} = \frac{1}{M} \sum_{j=1}^{M} \left[ \frac{1}{T_j} \sum_{t=1}^{T_j} \cos(f_{r,j,t}, f_{g,j,t}) \right] \tag{22}$$

where $M$ is the number of video pairs, $T_j$ is the frame count of the $j$-th video pair, and $f_{r,j,t}$, $f_{g,j,t}$ are the CLIP features of the $t$-th frame in the $j$-th video pair.

### B.3 Camera Trajectory Evaluation

**Absolute Trajectory Error (ATE).** Before evaluation, we align the estimated trajectory to the reference by a global Sim3 transform (scale, rotation, translation). Let the aligned pose components be $\tilde{\mathbf{t}}_{\text{est},i}$ and $\tilde{\mathbf{R}}_{\text{est},i}$. ATE measures global consistency by the Euclidean distance between corresponding camera positions:

$$\text{ATE}_i = \left\| \mathbf{t}_{\text{ref},i} - \tilde{\mathbf{t}}_{\text{est},i} \right\|_2, \qquad \text{ATE} = \sqrt{\frac{1}{n} \sum_{i=1}^{n} \text{ATE}_i^2}. \tag{23}$$

**Relative Pose Error — Translation (RPE-T).** RPE-T evaluates local translation accuracy between consecutive frames. Define relative motions via poses (index gap $\Delta = 1$):

$$\Delta\mathbf{T}_{\text{ref},i} = \mathbf{T}_{\text{ref},i}^{-1} \mathbf{T}_{\text{ref},i+1}, \qquad \Delta\mathbf{T}_{\text{est},i} = \tilde{\mathbf{T}}_{\text{est},i}^{-1} \tilde{\mathbf{T}}_{\text{est},i+1}. \tag{24}$$

Let $\Delta\mathbf{t}_{\text{ref},i}$ and $\Delta\mathbf{t}_{\text{est},i}$ be the translation parts of these relative transforms. The per-step error and RMSE are:

$$\text{RPE-T}_i = \left\| \Delta\mathbf{t}_{\text{ref},i} - \Delta\mathbf{t}_{\text{est},i} \right\|_2, \qquad \text{RPE-T} = \sqrt{\frac{1}{n-1} \sum_{i=1}^{n-1} \text{RPE-T}_i^2}. \tag{25}$$

**Relative Pose Error — Rotation (RPE-R).** RPE-R assesses the accuracy of orientation changes between consecutive frames. Let the relative rotations be

$$\Delta\mathbf{R}_{\text{ref},i} = \mathbf{R}_{\text{ref},i}^{-1} \mathbf{R}_{\text{ref},i+1}, \qquad \Delta\mathbf{R}_{\text{est},i} = \tilde{\mathbf{R}}_{\text{est},i}^{-1} \tilde{\mathbf{R}}_{\text{est},i+1}. \tag{26}$$

The per-step angular error (degrees) and RMSE are:

$$\text{RPE-R}_i = \arccos\left( \frac{\text{trace}(\Delta\mathbf{R}_{\text{ref},i}^\top \Delta\mathbf{R}_{\text{est},i}) - 1}{2} \right) \cdot \frac{180}{\pi}, \qquad \text{RPE-R} = \sqrt{\frac{1}{n-1} \sum_{i=1}^{n-1} \text{RPE-R}_i^2}. \tag{27}$$

### B.4 Implementation Details

**Preprocessing.** For FID, images are resized to $299 \times 299$ and fed to Inception-V3 with standard ImageNet normalization. For FVD and CLIP-based metrics, frames are resized to $224 \times 224$ with the respective model normalizations. For camera trajectory evaluation, images are resized to $720 \times 480$ and uniformly sampled to 20 frames while preserving the first and last frames. Videos are uniformly sampled to 25 frames with first/last preserved; for FVD, we further sample to 16 frames per clip; for CLIP-$\text{V}_{\text{sim}}$ on long videos, we cap processing at 20 frames.

**Evaluation Protocol.** For static scenes with multiple references, we estimate the real distribution from all reference images; for single-image scenes, we apply minimal augmentation to avoid singular covariances. Dynamic scenes maintain frame correspondence between generated and reference videos. For trajectories, poses are recovered by SfM, the estimated trajectory is aligned to the reference by Sim3 to resolve scale, and metrics are computed using `evo` with alignment and scale correction enabled.

# C DETAILS FOR FLF SCORING AND SETTINGS

This section provides a detailed breakdown of the Flow-Gated Latent Fusion (FLF) module, which is introduced in Section 3.3. The goal of FLF is to identify and selectively update latent channels that are highly relevant to motion, thereby preserving visual details encoded in appearance-focused channels. To achieve this, at each denoising step $i$, FLF computes a motion similarity score $S^{(t,c)}$ for each latent channel $c$. Below, we detail how this score is calculated.

**Optical Flow Computation** At each denoising step $i$, we compute optical flow maps for each channel $c$ of both the predicted latent $\hat{\mathbf{x}}_0^{(t)}$ and the target trajectory latent $\mathbf{Z}_{\text{traj}}$. The computation is performed frame-by-frame; that is, for each latent tensor, we calculate the dense optical flow between consecutive temporal frames using the Farnebäck algorithm (Farnebäck, 2003). This process yields a predicted flow map, $\mathcal{F}_{\text{pred}}^{(t,c)}$, and a ground-truth (GT) flow map, $\mathcal{F}_{\text{gt}}^{(t,c)}$. At each pixel, the flow is a 2D vector $(u_*, v_*)$ representing horizontal and vertical displacement. All subsequent metric calculations are performed over the set of valid (i.e., non-occluded) pixels, defined as $\Omega^{(t,c)} = \{(x, y, \tau) \mid \mathbf{M}^{(c)}(x, y, \tau) = 1\}$, where $(x, y)$ are pixel coordinates and $\tau$ is the frame index. Since optical flow is computed between adjacent frames, for a latent tensor with $T_l$ total frames, the index $\tau$ ranges from 1 to $T_l - 1$.

**Metric Calculation** The motion score $S^{(t,c)}$ is derived from three standard optical flow metrics that quantify the error between the predicted flow $\mathcal{F}_{\text{pred}}^{(t,c)}$ and the ground-truth flow $\mathcal{F}_{\text{gt}}^{(t,c)}$ at each step $i$.

- **Masked End-point Error (M-EPE)** measures the average Euclidean distance between the predicted and GT flow vectors over all valid pixels:

$$\text{M-EPE}^{(t,c)} = \frac{1}{|\Omega^{(t,c)}|} \sum_{(x,y,\tau) \in \Omega^{(t,c)}} \left\| \mathcal{F}_{\text{pred}}^{(t,c)}(x, y, \tau) - \mathcal{F}_{\text{gt}}^{(t,c)}(x, y, \tau) \right\|_2. \tag{28}$$

- **Masked Angular Error (M-AE)** calculates the average angular difference in radians between the flow vectors:

$$\text{M-AE}^{(t,c)} = \frac{1}{|\Omega^{(t,c)}|} \sum_{(x,y,\tau) \in \Omega^{(t,c)}} \arccos \left( \frac{\mathcal{F}_{\text{pred}}^{(t,c)}(x, y, \tau) \cdot \mathcal{F}_{\text{gt}}^{(t,c)}(x, y, \tau)}{\|\mathcal{F}_{\text{pred}}^{(t,c)}(x, y, \tau)\| \cdot \|\mathcal{F}_{\text{gt}}^{(t,c)}(x, y, \tau)\|} \right). \tag{29}$$

- **Outlier Percentage (Fl-all)** is the percentage of pixels in $\Omega^{(t,c)}$ where the flow estimation is considered erroneous. Following standard benchmarks, a pixel is flagged as an outlier if its M-EPE exceeds 3 pixels or if its relative error, $\|\mathcal{F}_{\text{pred}}^{(t,c)} - \mathcal{F}_{\text{gt}}^{(t,c)}\|_2 / \|\mathcal{F}_{\text{gt}}^{(t,c)}\|_2$, is greater than 5%. We denote this outlier percentage as $\text{F}^{(t,c)}$.

**Normalization and Weighting** The three metrics exist on different scales, so we first normalize each to the range $[0, 1]$ before combining them. This corresponds to the $\text{Norm}_k^{(t,c)}$ terms used in the main text:

$$\text{Norm}_{\text{E}}^{(t,c)} = \min(\text{M-EPE}^{(t,c)}/n_{\text{E}}, 1),$$
$$\text{Norm}_{\text{A}}^{(t,c)} = \min(\text{M-AE}^{(t,c)}/n_{\text{A}}, 1), \tag{30}$$
$$\text{Norm}_{\text{F}}^{(t,c)} = \min(\text{F}^{(t,c)}/n_{\text{F}}, 1),$$

where $n_{\text{E}}, n_{\text{A}}$, and $n_{\text{F}}$ are normalization constants chosen to reflect typical value ranges for each metric. The final motion score $S^{(t,c)}$ is a weighted sum of the inverted normalized errors, as defined

in Eq. (6) in the main text. The weights $\gamma_k$ (where $k \in \{E, A, F\}$ and $\sum_k \gamma_k = 1$) and the normalization constants are set based on common practices in optical flow evaluation to balance each metric's contribution. In our experiments, we set $n_E = 10$, $n_A = 30$, and $n_F = 0.5$. The weights in Eq. (6) are set to $(\gamma_E, \gamma_A, \gamma_F) = (0.4, 0.3, 0.3)$.

## D  VISUAL COMPARISON OF DSG AND NAIVE CFG

In contrast, our proposed DSG mechanism successfully balances trajectory control and visual fidelity. The images generated with DSG maintain structural integrity and high perceptual quality while closely following the intended camera path. This comparison validates our central claim that a specialized guidance mechanism like DSG is necessary for robust control in our warp-and-repaint framework and demonstrates its clear superiority over a direct application of CFG.

As discussed in the main text, our Dual-Path Self-Corrective Guidance (DSG) is specifically designed to handle the large angular difference between the guided velocity ($\mathbf{v}_t^{\text{traj}}$) and the unguided velocity ($\mathbf{v}_t^{\text{ori}}$). This large divergence, with observed cosine similarities often between 0.3–0.6, renders a naive application of the standard Classifier-Free Guidance (CFG) formula ineffective, as CFG is designed for scenarios where the conditional and unconditional paths are closely aligned.

To visually demonstrate this, we conduct a direct comparison, shown in Fig. 5. We compare our full DSG framework against both a standard CFG implementation and an ablated version of our method without the adaptive weight $\beta_t$. As the results show, applying a naive CFG formulation leads to severe visual artifacts and distorted object structures, while removing the adaptive weight also reduces guidance stability.

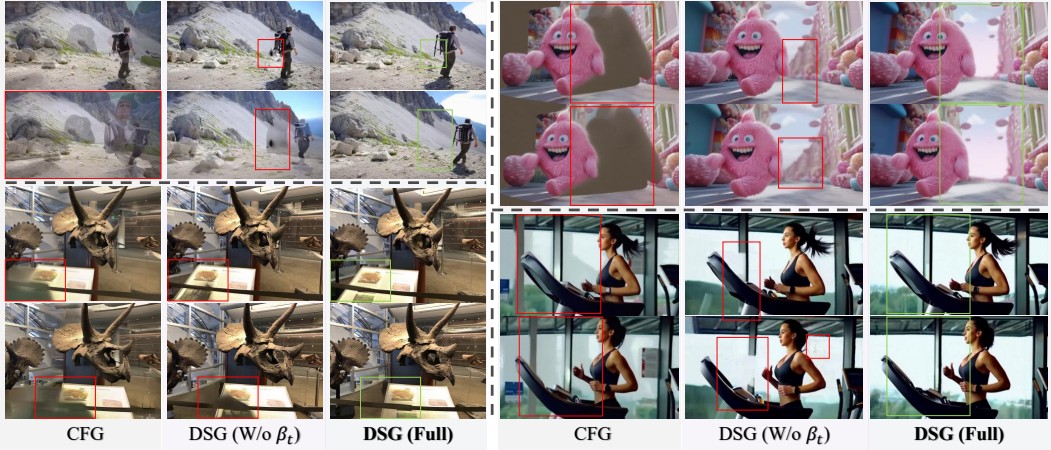

Figure 5: Visual comparison of our proposed DSG and naive CFG for trajectory guidance. In many test scenes, naive CFG fails to handle the large angular difference between the two velocity fields, resulting in significant artifacts and errors. Removing the adaptive weighting factor $\beta_t$ from our method (**DSG w/o** $\beta_t$) leads to reduced guidance stability and introduces errors. In contrast, our full DSG framework stably generates high-fidelity and structurally consistent results.

## E  MORE EXPERIMENTAL RESULTS

This section provides additional experiments and results that complement the findings presented in the main paper. We include a detailed efficiency analysis, further ablation studies, and more qualitative examples to fully demonstrate the capabilities and robustness of our framework.

### E.1 Efficiency and Runtime Analysis

As mentioned in Section 4.1, our framework is training-free and operates entirely at inference time. Table 2 provides a detailed comparison of inference efficiency against several state-of-the-art methods on a single NVIDIA A100 GPU.

Our method incurs zero training cost, which is a significant advantage over approaches that require costly fine-tuning. The primary computational overhead comes from the Intra-Step Recursive Refinement (IRR) module, which increases the inference time by approximately 40–50% compared to running only the base VDM. Despite this, our framework achieves inference speeds that are comparable to, and in some cases faster than, prior methods. For instance, when integrated with the SVD backbone, our method is more efficient than several other SVD-based baselines. This analysis demonstrates that our framework achieves strong controllability without prohibitive computational costs, offering an efficient and effective alternative to training-intensive pipelines.

Table 2: Efficiency comparison. We measure inference time on a single NVIDIA A100 across methods built on SVD (Blattmann et al., 2023), Wan 2.1(Wan et al., 2025), CogVideoX (Yang et al., 2024), and custom backbones. ReCamMaster (Bai et al., 2025a) is evaluated at 81 frames; all others use 25 frames. Our method is training-free and plug-and-play, thus incurring zero training cost. Its runtime adds  40% over the base video model, attributable to the IRR recursive refinement. Overall, it achieves comparable or faster inference than prior approaches while avoiding any training overhead.

|  | Frames | Resolution | Inference Time (min) | Base Video Model | Training-Free |
|---|---|---|---|---|---|
| See3D (Ma et al., 2025a) | 25 | $576 \times 1024$ | 1.7 | Custom | ✗ |
| ViewCrafter (Yu et al., 2024c) | 25 | $576 \times 1024$ | 1.8 | Custom | ✗ |
| ViewExtrapolator (Liu et al., 2024) | 25 | $576 \times 1024$ | 1.6 | SVD | ✓ |
| TrajectoryAttention (Xiao et al., 2025) | 25 | $576 \times 1024$ | 5.5 | SVD | ✗ |
| TrajectoryCrafter (Yu et al., 2025) | 25 | $384 \times 672$ | 1.7 | CogVideoX | ✗ |
| NVS-Solver (You et al., 2025) | 25 | $576 \times 1024$ | 9.3 | SVD | ✓ |
| ReCamMaster (Bai et al., 2025a) | 81 | $480 \times 832$ | 14.6 | Wan 2.1 T2V | ✗ |
| WorldForge (Ours, 720P) | 25 | $720 \times 1280$ | 17.3 | Wan 2.1 I2V | ✓ |
| WorldForge (Ours, 480P) | 25 | $480 \times 832$ | 6.8 | Wan 2.1 I2V | ✓ |
| WorldForge (Ours, on SVD) | 25 | $576 \times 1024$ | 1.3 | SVD | ✓ |

### E.2 Ablation on Video Diffusion Models

To verify that our performance is due to our proposed guidance mechanism and not just the power of the primary VDM (Wan2.1), we conducted an ablation study by porting our entire framework to the widely-used, U-Net-based Stable Video Diffusion (SVD) model (Blattmann et al., 2023). We made a minor adjustment to adapt to the characteristics of SVD and its native EDM sample. We then performed a fair comparison against other state-of-the-art methods that are also built on the SVD backbone, using identical inputs.

The results, shown in Fig. 6, demonstrate that our guidance transfers seamlessly to this new architecture. It makes the native SVD model controllable and capable of following specific trajectories, achieving SOTA performance among SVD-based methods in content quality, structural plausibility, and trajectory consistency. This experiment confirms that our guidance is architecture-independent and suggests it will have strong potential when paired with future, more powerful base models.

### E.3 Ablation on Depth Estimation Models

Our framework relies on a warp-and-repaint strategy, where the quality of the initial warp is dependent on a depth estimation model. To test the robustness and flexibility of our approach, we evaluated its performance with several different state-of-the-art depth estimators: VGGT (Wang et al., 2025b), UniDepth (Piccinelli et al., 2024), Mega-SaM (Li et al., 2025), and DepthCrafter (Hu et al., 2025).

As shown in Fig. 7, our method demonstrates broad compatibility and maintains high performance across all tested depth models. Even when the depth-based warping produces challenging inputs with noise, errors, or significant missing regions (disocclusions), our framework effectively compensates. This resilience stems from the strong generative world priors of the underlying VDM,

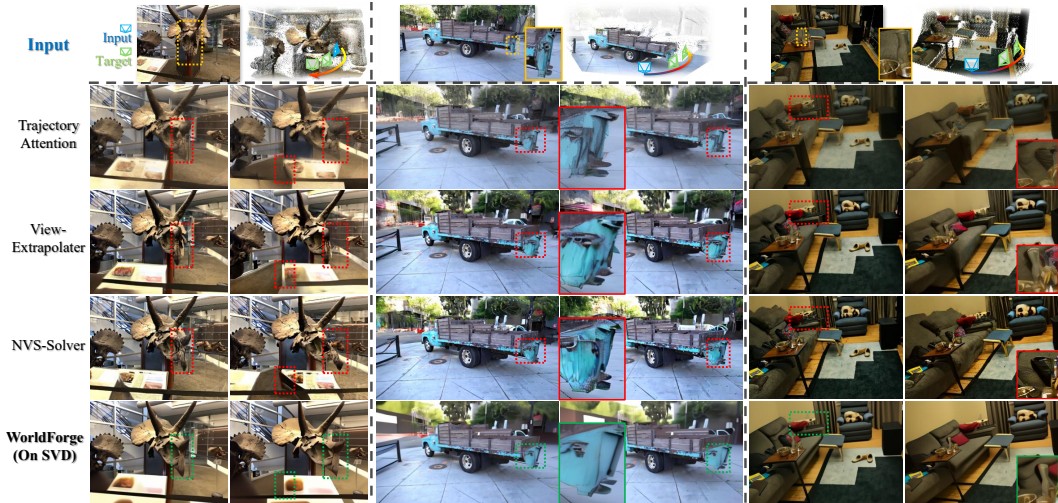

Figure 6: Ablation across different VDMs. To rule out the influence of the intrinsic performance advantage of the VDM (Wan2.1 (Wan et al., 2025)) and to verify the method's transferability, we port the proposed guidance to a compact U-Net–based SVD model (Blattmann et al., 2023) and compare against SVD-based SOTA baselines. Experiments show that the guidance transfers seamlessly, makes the native SVD controllable, and achieves SOTA performance in content quality, structural plausibility, and trajectory consistency.

which are leveraged by our guidance modules to correct artifacts and plausibly fill in missing areas during the repainting stage. This self-correction capability confirms that our framework can be used in a plug-and-play manner with various depth estimation techniques without sacrificing the quality of the final output.

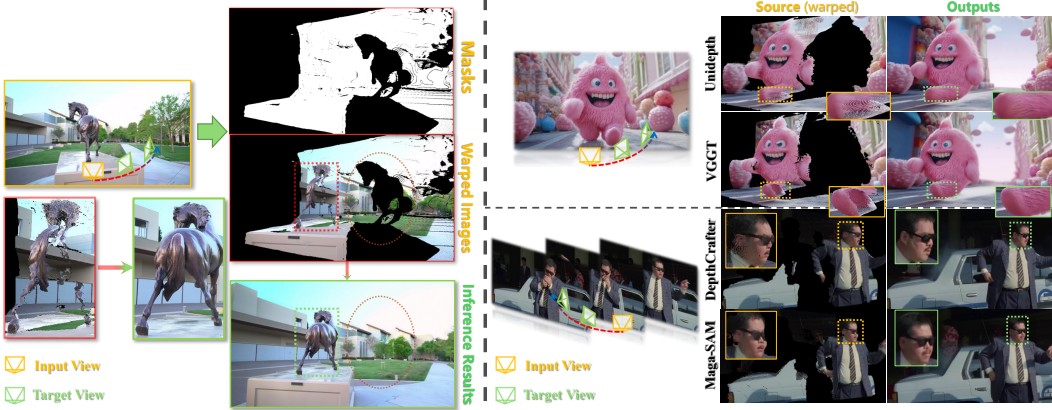

Figure 7: Depth-models ablation. Our method leverages the inherent world knowledge of VDMs to correct errors and fill missing regions even under challenging inputs (left). This strong self-correction ability ensures broad compatibility with different depth estimators (right). Despite variations or noise in depth-based warping, it reliably compensates through learned priors and produces realistic, high-quality results.

### E.4 APPLICATIONS IN VIDEO EDITING

Beyond trajectory-controlled generation, our framework's flexibility makes it a powerful tool for various video post-production and editing tasks. This includes effects like video stabilization, camera freezing, and dynamic viewpoint switching.

Furthermore, by incorporating a flexible masking strategy, our framework can perform diverse content edits such as object removal, addition, subject replacement, and virtual try-on seamlessly. The general process for these edits involves first segmenting the target region in each frame using a tool like SAM (Kirillov et al., 2023). The desired edit is then applied to the first frame (e.g., using Gemini (Comanici et al., 2025)). Finally, this edited frame and the corresponding masks are processed by our pipeline to render a temporally consistent result. For adding new objects where none exist in the source video, a simple bounding box can be provided to guide the placement. Fig. 8 shows several qualitative examples of these video editing effects.

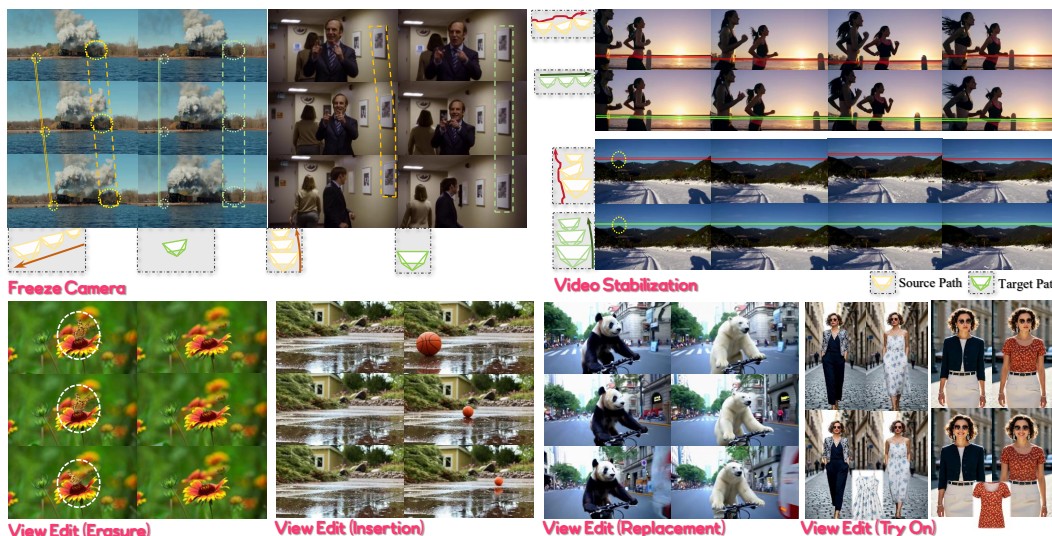

Figure 8: Other video effects enabled by our method. Beyond video re-cam, our flexible depth-based warping also supports various video editing operations, such as freezing the camera, stabilizing video, and editing video content. These extensions further broaden the practical scope of our approach.

### E.5 GENERATION ON CHALLENGING SCENES

Our approach demonstrates robust performance in difficult cases where other methods may falter. We highlight two such scenarios: human-centric scenes and single-image 360° view generation.

**Human-Centric Scenes** Human-centric scenes are challenging for novel view synthesis due to the need for high structural and temporal consistency. As shown in Fig. 9, some methods can struggle with these cases, sometimes introducing artifacts, unintended motion, or difficulty rendering plausible facial features. For instance, TrajectoryCrafter (Yu et al., 2025) may recover the coarse structure, but can introduce unnatural facial deformations. In contrast, our method's use of strong generative priors and precise trajectory guidance helps maintain scene stationarity and consistency, producing more natural renderings that better preserve the subject's appearance.

**360° View Generation** Generating full 360° views from a single image is another demanding task. Our framework's precise trajectory control helps address the limited field-of-view of the source image, enabling the creation of coherent, object-centric orbit views of complex scenes, as shown in Fig. 10. We achieve the full 360° loop by generating a sequence where the final frame seamlessly connects to the first. This is made possible by our precise guidance, which maintains high image

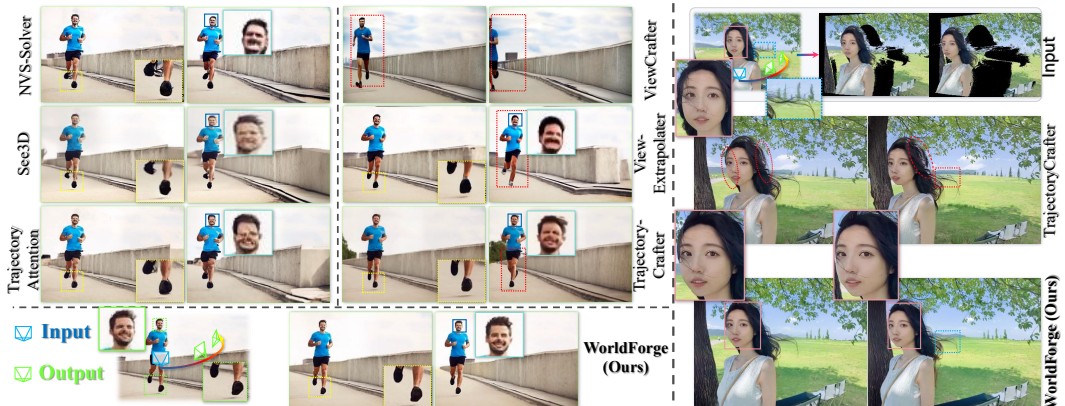

Figure 9: Static 3D generation on human-centric scenes. Existing methods struggle, particularly with motion-prone shots (left) and portrait close-ups (right). On the left, baselines introduce artifacts and unintended motion. On the right, most fail to produce plausible results; TrajectoryCrafter (Yu et al., 2025) recovers coarse structure but lacks detail and visual appeal. In contrast, our method maintains scene stationarity under trajectory guidance and produces natural, faithful renderings, achieving both precise control and high perceptual quality.

quality and prevents the accumulation of errors over the entire long-range trajectory, a common point of failure for other methods. Unlike traditional panoramic approaches, our method directly generates a continuous view along a given trajectory. This can offer more flexibility and strong visual quality, particularly for object-centric paths.

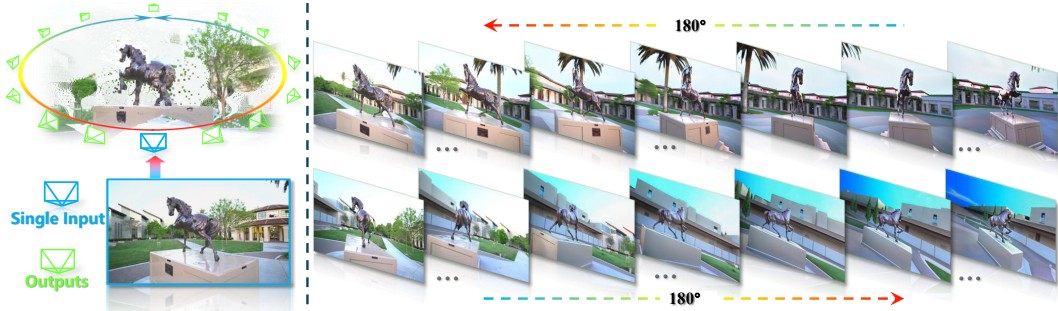

Figure 10: 360° orbit views from a single real-world outdoor image. With precise trajectory control and realistic rendering, our method overcomes the viewpoint limitation of single-image generation and produces ultra-wide views of complex real scenes. Unlike panorama-based approaches, it directly supports object-centric trajectories and achieves higher visual quality.

## USE OF LARGE LANGUAGE MODELS (LLMS)

A Large Language Model (LLM) was used as a writing assistant in the preparation of this manuscript. Its primary role was to aid in polishing the language, improving the clarity of the text, and generating descriptive text for figures. This included tasks such as rephrasing sentences for conciseness, correcting grammatical errors, and ensuring a consistent academic tone. All authors have reviewed and edited the LLM-generated suggestions and take full responsibility for the scientific accuracy and final content of this paper.

