# OpenReview forum: "WorldForge: Unlocking Emergent 3D/4D Generation in Video Diffusion Model via Training-Free Guidance"
_ICLR.cc/2026/Conference — ICLR 2026 Conference Withdrawn Submission_

### Official Review · Reviewer_M2Gj · 2025-10-19

**Soundness:** 1
**Presentation:** 2
**Contribution:** 2
**Rating:** 2
**Confidence:** 4

**Summary:**

The paper introduces WorldForge, a training-free, inference-time framework that addresses limited controllability and entangled scene–camera dynamics in video diffusion models (VDMs).

The method warps input frames along a reference trajectory and repaints them using a pre-trained VDM, combining three complementary mechanisms for trajectory-controlled generation.
(1) Intra-Step Recursive Refinement: fine-grained, step-wise trajectory guidance
(2) Flow-Gated Latent Fusion: decouple motion from appearance via flow similarity and inject guidance into motion-related channels
(3) Dual-Path Self-Corrective: Guidance that contrasts guided vs. unguided paths to correct trajectory drift.

Experiments report state-of-the-art results in trajectory adherence and perceptual quality, outperforming both training-heavy and inference-only baselines.

**Strengths:**

- The proposed method is a training-free framework that leverages pre-trained video diffusion models, exploiting the high visual fidelity of video models.
- The proposed method is adaptable across different video diffusion backbones and trajectory-control tasks.

**Weaknesses:**

**Weakness**
1. Evaluation credibility and transparency.\
(1) Table 1 data sources: The claim of evaluating single-view 3D scene generation on “diverse internet, real-world, and AI-generated images” lacks concrete dataset names, counts, and acquisition criteria.\
(2) Ablation, Backbone and guidance comparisons: Ablation, replacing the base video diffusion model, and comparing CFG vs. DSG, are shown mainly with qualitative figures; quantitative metrics and protocols are needed to establish clear superiority.\
(3) Sec. 4.2 claims a comparison with ReCamMaster, but Table 1 does not report it.

2. Hyperparameter Ambiguity .\
Several hyperparameters are introduced (e.g., λ used for defining δ in Eq. 7, ρ in Eq. 8), but their impact is not explored via ablations or sensitivity studies.\
Also, when defining δ in Eq. (7), the exact value/schedule of λ is not specified, making it difficult to reproduce or interpret the behavior across timesteps.


**Minor Suggestions:**
1. Related work positioning.
(1) While β is defined differently in Eq. (8), the idea of decomposing into parallel/orthogonal components and reinforcing the orthogonal part echoes [1]APG. Citing and clearly contrasting the shared ideas, key differences, and the advantages of this formulation for trajectory-controlled generation would improve the paper’s quality.

2. Figure Size.\
Figure 3 is difficult to read at the current size, even when zoomed.

3. Terminology.\
The term “3D scene generation” seems potentially misleading. The task appears closer to single-image novel view synthesis rather than full 3D representation learning/generation.

[1] ICLR 2024, APG: Eliminating Oversaturation and Artifacts of High Guidance Scales in Diffusion Models

**Questions:**

1. Relation to [1]RePaint.\
The pipeline resembles RePaint in spirit. My understanding is that your method differs in whether masked fusion uses clean vs. noisy inputs (and when noise is injected). If you apply noised inputs to the masked fuse step, how does behavior change compared to the clean-then-noise strategy?

2. VAE channel roles.\
Do you have references showing that the VAE channels separate into appearance-focused vs. motion-focused components? Is “appearance” defined mainly by not encoding motion? Clarification (and citations) would help.
And regarding the “appearance-focused channels”, if some channels capture motion, is it more accurate to say they retain position-aware appearance across frames, while other channels do not preserve such spatial information? A short analysis (e.g., per-channel flow/stability) would be informative.

3. Understanding Eq. (7).\
At t = T (pure noise), it is unclear why the motion similarity score would be high anywhere. It seems λ must be large in the early stage so that masked latent fusion can seed regions that subsequently yield higher motion-similarity scores. How is λ scheduled/defined over timesteps, and what is the authors’ intuition over this equation? For instance, only in regions where projection is meaningful, the score could adopt generation when motion is not reflected—neglecting the projection latent. Is that the intended behavior?

4. Mask application schedule.\
Is masked latent fuse applied at every timestep, or only within a window (e.g., early/mid/late denoising steps)?

5. Comparision with ReCamMaster[2]\
Has a comparison with ReCamMaster been conducted? If so, quantitative evaluation results would be valuable.

[1] CVPR 2022, RePaint: Inpainting using Denoising Diffusion Probabilistic Models
[2] ICCV 2025, ReCamMaster: Camera-Controlled Generative Rendering from A Single Video

---

### Official Review · Reviewer_GEw5 · 2025-10-26

**Soundness:** 3
**Presentation:** 3
**Contribution:** 2
**Rating:** 6
**Confidence:** 3

**Summary:**

WorldForge is a training-free, inference-time guidance framework that steers a pre-trained video diffusion model (VDM) for controllable 3D/4D generation (single-image 3D NVS and 4D trajectory-controlled re-rendering). It introduces three coupled mechanisms—Intra-Step Recursive Refinement (IRR) to inject trajectory cues inside each denoising step, Flow-Gated Latent Fusion (FLF) to selectively fuse warp-based trajectory latents into motion-relevant VAE channels, and Dual-Path Self-Corrective Guidance (DSG) to reconcile a guided vs. unguided denoising path via an adaptive correction term. The pipeline follows a warp-and-repaint design using monocular depth/pose to render a trajectory-conditioned “guidance video,” then applies IRR/FLF/DSG during sampling of backbones such as Wan2.1 or SVD; experiments report state-of-the-art results across 3D/4D tasks (e.g., FID/FVD and ATE/RPE), with ~40–50% extra inference cost due to IRR but no training cost.

**Strengths:**

1. Clear, training-free formulation for precise trajectory control. The paper squarely targets the VDM gaps and proposes a plug-and-play guidance suite that preserves base-model priors while injecting control.

2. DSG is an adaptive, self-referential correction beyond CFG. Calibrating the correction by cosine similarity between guided/unguided velocities (and using the orthogonal component when they diverge) is a well-argued upgrade over naive CFG blending for trajectory guidance.

3. Single-image 3D NVS and 4D re-rendering use the same recipe. The framework is model-agnostic (Wan2.1/SVD) and depth-agnostic.

**Weaknesses:**

1. Core pipeline still depends on warp-and-repaint with monocular depth—failure modes remain under-analyzed. The paper acknowledges depth/occlusion errors can corrupt guidance, but quantitative stress-tests for noisy/inaccurate warps (e.g., heavy parallax, reflective scenes, thin structures) are limited; DSG/FLF’s robustness under severe OOD warps isn’t systematically evaluated.

2. Evidence for true 3D consistency is mostly proxy-level. ATE/RPE are camera-trajectory metrics; there is no explicit metric for volumetric/geometry consistency (e.g., multiview reprojection error, dynamic point-cloud completeness) to ensure that scene structure doesn’t “swim” across views.

3. “Model-agnostic” claim would benefit from deeper SVD results. SVD experiments are mentioned and positioned as positive, but the main results lean heavily on Wan2.1; more equal-footing SVD numbers/visuals would reduce the risk that gains hinge on a stronger base prior.

4. Compute/latency overhead is only summarized. A stated 40–50% inference slowdown (due to per-step IRR) is non-trivial for long videos or high-res outputs; a time-budget analysis across lengths/resolutions and a wall-clock comparison against tuned repainting baselines would clarify practical impact.

**Questions:**

1. Robustness to depth noise: Under deliberately corrupted depth (e.g., scale-shift, flipped planes, 10–30% random occlusion mask errors), how do ATE/RPE and FVD change, and how much do IRR/FLF/DSG respectively recover?

2. 3D consistency metrics: Beyond camera errors, can you report multiview photometric error or dynamic point-cloud metrics (e.g., chamfer/complete rate) across rendered trajectories to substantiate “geometric consistency” claims?

3. FLF channel-selection details and cost: How is optical flow computed per latent channel in practice (estimator, resolution, compute), and how does the dynamic threshold scale with video length/resolution?

4. Backbone/model-agnosticism evidence: Provide fuller SVD tables/visuals (same datasets/metrics as Table 1) to demonstrate that gains are not Wan-specific, and include runtime/memory figures per backbone.

---

### Official Review · Reviewer_Xusz · 2025-10-31

**Soundness:** 2
**Presentation:** 3
**Contribution:** 2
**Rating:** 2
**Confidence:** 4

**Summary:**

This paper introduces WorldForge, a training-free, inference-time framework designed to enhance controllability and spatio-temporal consistency in video diffusion models.Its core contributions are three tightly integrated modules: Intra-Step Recursive Refinement – injects fine-grained trajectory guidance at each denoising step through a recursive correction loop, ensuring motion stays aligned with the target path. Flow-Gated Latent Fusion – uses optical flow similarity to decouple motion and appearance in the latent space, selectively applying trajectory guidance only to motion-related channels. Dual-Path Self-Corrective Guidance – compares guided and unguided denoising paths to adaptively correct trajectory drift caused by noisy or misaligned structural signals.

**Strengths:**

1. The experimental results are godd, and the video quality looks very high.

2. There are many potential downstream applications.

**Weaknesses:**

1. The motivation for introducing Flow-Gated Latent Fusion (FLF) is quite unclear. After carefully checking, I only found one brief mention — “some channels specializing in appearance and others in motion” — which leaves readers confused. To make sense of this, I had to rely on prior knowledge, such as from “Video Diffusion Models are Training-free Motion Interpreter and Controller (NeurIPS 2024)”, which draws a similar conclusion and uses a comparable approach — selectively replacing motion-related channels based on motion scores. Without a clear explanation of the motivation and proper citation, I find it difficult to accept this component as justified.


2. I didn’t fully understand Section 3.4 (around line 302). Do you mean that in Wan, $V_{con}$ and $V_{uncon}$ are supposed to be close to 1, but in the warping examples you measured 0.3–0.6, so you increased the distance accordingly? Please correct me if this interpretation is wrong. Based on this understanding, I have some questions. In which part of the Wan's technical report is it mentioned? Is this conclusion specific to Wan, or would it still hold for other models such as SVD, CogVideoX, or future architectures? If it generalizes, what’s the rationale — is it simply empirical tuning derived from Wan’s behavior?


3. The Introduction writing needs improvement. Lines 68–78 provide a good summary of the limitations of prior methods, but the next three paragraphs abruptly introduce many technical terms and details without explaining why each is necessary. For example, in lines 71–75, you mention that prior warping-based methods suffer from OOD issues — I expected a clear follow-up describing how your approach addresses this problem, but I couldn’t find keywords or sections explicitly tackling it. Was this issue resolved through the DSG component?


4. Regarding the four listed contributions, the method itself is not a training-free trajectory control approach. For IRR, nearly all repainting methods already rely on masks for inpainting. As for FLF and DSG, as mentioned above, their motivations remain unclear and insufficiently explained.


5. For the evaluation metrics, the datasets used all contain camera pose information, making it possible to compute pixel-level metrics such as PSNR and SSIM. Why, then, were only FVD and CLIP-based metrics reported, which are less sensitive to geometric and pixel-level consistency?

**Questions:**

1. What is the definition of  $Z_{traj}$.I got lost around line 246 — there’s no clear definition provided, which makes it hard to fully understand Equation (5). Does it refer to warping frames?

2.Compared with previous methods that tend to produce artifacts, your approach achieves clear improvements.
What do you think is the most critical component that contributes to this advancement?

---

### Official Review · Reviewer_RWms · 2025-11-01

**Soundness:** 2
**Presentation:** 2
**Contribution:** 2
**Rating:** 4
**Confidence:** 4

**Summary:**

The paper presents WorldForge, a training-free inference-time framework for controllable 3D/4D generation using pre-trained video diffusion models. It introduces three modules, Intra-Step Recursive Refinement (IRR), Flow-Gated Latent Fusion (FLF), and Dual-Path Self-Corrective Guidance (DSG), to inject trajectory-aligned guidance without additional training. The method aims to improve spatial consistency and camera controllability in 3D/4D video synthesis.

**Strengths:**

- Training-free design that avoids costly fine-tuning and preserves pretrained priors.
- Conceptually interesting inference-time guidance that improves trajectory adherence and geometric stability.
- Qualitative results show better spatial-temporal consistency than prior training-free baselines.

**Weaknesses:**

- The motivation and method are misaligned: the paper claims to “unlock emergent 3D/4D generation,” but the contribution is limited to trajectory correction rather than broader world-modeling capability.
- The main quantitative comparison is not fair: the proposed method uses a large-scale Wan 14B backbone, while most baselines rely on smaller or older models. This makes it difficult to isolate the gain from the proposed modules versus model scale. To compensate for this, further quantitative comparison on SVD-based baselines, which is in Fig.6 , will be needed.
- No component-wise quantitative ablation is provided; the effect of IRR, FLF, and DSG is demonstrated only qualitatively.
- Inference efficiency is poor: inference time increases by roughly 40–50 % and requires additional external depth or optical-flow models.
- Figures and mathematical exposition are dense, making the method difficult to follow.
- Despite the claim of model-agnosticism, quantitative results on other backbones (e.g., CogVideoX, SVD) are missing.

**Questions:**

- Are there latent channels that consistently exhibit high motion-similarity scores in FLF? Some statistical or visual analysis would clarify its behavior.
- When applied to smaller models like SVD, the outputs appear blurrier, quantitative comparisons or explanations for this degradation would strengthen the claim of generality.

---

### Note · Authors · 2025-11-13

I have read and agree with the venue's withdrawal policy on behalf of myself and my co-authors.